# Suppression of hollow droplet rebound on super-repellent surfaces

Ying Zhou [1,4], Chenguang Zhang[2,4], Wenchang Zhao[1], Shiyu Wang [1] & Pingan Zhu [1,3] ✉

Droplet rebound is ubiquitous on super-repellent surfaces. Conversion between kinetic and surface energies suggests that rebound suppression is unachievable due to negligible energy dissipation. Here, we present an effective approach to suppressing rebounds by incorporating bubbles into droplets, even in super-repellent states. This suppression arises from the counteractive capillary effects within bubble-encapsulated hollow droplets. The capillary flows induced by the deformed inner-bubble surface counterbalance those driven by the outer-droplet surface, resulting in a reduction of the effective take-off momentum. We propose a double-spring system with reduced effective elasticity for hollow droplets, wherein the competing springs offer distinct behavior from the classical single-spring model employed for single-phase droplets. Through experimental, analytical, and numerical validations, we establish a comprehensive and unified understanding of droplet rebound, by which the behavior of single-phase droplets represents the exceptional case of zero bubble volume and can be encompassed within this overarching framework.

Liquid droplets bounce deftly when they impact super-repellent surfaces, exhibiting remarkable elasticity owing to the efficient conversion between kinetic and surface energies[1-6]. Previous attempts to prevent droplet rebound rely on augmenting energy dissipation, for example, by using sticky surfaces[7,8] for enhanced liquid-solid adhesion, designing compound drops[9-11] to form a self-lubricating liquid layer for droplet arrest, elevating the temperature of water[12] to produce adhesive droplets through reduced surface tension and vapor condensation, and introducing droplet additives such as polymers[13,14] to increase the non-Newtonian elongational viscosity, surfactants[15,16] to decrease the surface tension, and oppositely charged polyelectrolytes[17] to produce hydrophilic surface defects. However, wetting transition occurs in these systems and unduly compromises liquid repellency.

With negligible energy dissipation, a bouncing droplet is analogous to a spring[2,18,19]. The solid-liquid contact time $t_c$ scales with the natural oscillation period as, $t_c \sim (m/\gamma)^{1/2} \sim (\rho R^3/\gamma)^{1/2}$, where $m$, $\gamma$, $\rho$, and $R$ represent the mass, surface tension, volumetric density, and radius of the droplet, respectively. The contact time of a millimetric droplet impacting super-repellent surfaces is of the order of several milliseconds. In contrast, non-rebound droplets imply an infinitely large contact time $t_c$, which requires either infinite mass or vanishing surface tension in the mass–spring model. Heavy droplets cannot bounce off solid surfaces when $R$ is larger than the capillary length, $l_c \sim (\gamma/\rho g)^{1/2}$ with $g$ the gravitational acceleration, because gravity dominates capillary effects[18]. However, the non-rebound behavior of capillary-dominant droplets ($R < l_c$) on super-repellent surfaces remains a challenge, as it is inherently impossible to achieve a surface tension of zero for any liquid.

Hollow droplets, which consist of a gas bubble surrounded by a liquid shell, possess a distinct core-shell structure that imparts them with unique physical and chemical properties compared to single-phase droplets. This distinctive morphology is prevalent in various practical processes[20-23], including raindrops colliding with the Earth's surface[20], deposition of hollow spherical particles during

[1]Department of Mechanical Engineering, City University of Hong Kong, 999077 Hong Kong, China. [2]Ansys Inc., 10 Cavendish Ct, Lebanon, NH 03766, USA. [3]Shenzhen Research Institute, City University of Hong Kong, 518057 Shenzhen, China. [4]These authors contributed equally: Ying Zhou, Chenguang Zhang. ✉e-mail: pingazhu@cityu.edu.hk

thermal spray coating[24], bubble-bursting in the preparation of nanoemulsions[25], and aerosol transfer from sea[26]. Prior research has contributed to our understanding of the dynamic behavior exhibited by hollow droplets, such as shape oscillations[27], droplet spreading[21,28–32], and counter-jet formation[21,28–32] during impact. However, previous studies have predominantly focused on the interactions of hollow droplets with hydrophilic or wetting solid surfaces[23,28,30–32], as well as liquid pools[33]. In contrast, there has been limited exploration regarding the impact of hollow droplets on nonwetting surfaces[31].

Here we present evidence of the suppressed rebound of hollow droplets upon impact with super-repellent surfaces, a phenomenon that cannot be achieved with single-phase droplets. The ability to suppress droplet rebound presents exciting prospects for the advancement of droplet-based shock absorbers, which can facilitate non-sticking liquid deposition, offering practical implications in various fields such as spray cooling[34,35], self-cleaning[36,37], inkjet printing[38,39], agricultural spraying[13,14], liquid transport[40,41], and fire extinguishing[42,43].

## Results

### Non-rebound of hollow droplets

To distinguish the bouncing dynamics of a single-phase droplet (SD) and hollow droplet (HD), we released them from the same height $H_0$ with identical impacting velocities $V_0 = (2gH_0)^{1/2}$ and kinetic energies. To ensure a fair comparison, the mass and liquid material of the two droplets were the same. Consequently, both droplets had the same liquid volume and characteristic radius $R_h = (R_0^3 - R_b^3)^{1/3}$, where $R_0$ and $R_b$ denote the apparent radius of the droplet (Fig. 1a, b) and radius of the encapsulated bubble, respectively (Fig. 1b). The volume fraction of the bubble was determined as $\Phi = R_b^3/R_0^3$. Notably, SDs can be considered a special instance of HDs with a completely filled core ($\Phi = 0$) and, consequently, the two radii are identical, $R_h = R_0$. HDs were generated and controlled in a co-flow microfluidic device (Supplementary Fig. 1), in which air and water were used as the core and shell fluids, respectively. Sodium dodecyl benzene sulfonate (SDBS) was added to water as the surfactant and the concentration was adjusted to stabilize HDs, by which the surface tension was in the range of 35–53 mN m$^{-1}$ (see details in "Methods"). To ensure a negligible effect of gravity, we kept $R_h < l_c$ in experiments.

Figure 1c, d show the impact of an SD and HD on a superhydrophobic surface (Supplementary Fig. 2), respectively. Unlike the SD, which bounces off the surface (Fig. 1c), the HD wobbles and promptly rests on the surface after collision (Fig. 1d and Supplementary Movie 1). Compared with the SD, the HD exhibits a considerably smaller oscillating magnitude in the position of its centroid and the aspect ratio of its shape (Supplementary Fig. 3). The difference in the dynamics can be further illustrated by considering the temporal evolution of the droplet's bottom height $H_b$. Figure 1e shows that the bottom height of the HD remains zero after impact, whereas that of the SD increases to be higher than zero to form multiple positive peaks owing to the rebound. Considering that non-rebound HDs remain in contact with the solid surface after impingement, we can assume that the contact time would tend towards infinity when compared to the bouncing case, neglecting droplet evaporation. In experiments, a transition from rebound to non-rebound behavior occurred at a moderate value of the bubble volume fraction ($\Phi \approx 0.4$), as indicated by a sharp increase in contact time (Fig. 1f). The HD-enabled rebound suppression is ubiquitous, irrespective of the droplets and super-repellent surfaces, such as air-in-hexadecane HDs on super-amphiphobic surfaces, air-in-water HDs on superhydrophobic surfaces, air-in-SDBS HDs on horizontal and inclined surfaces in Leidenfrost regime, air-in-SDBS HDs impacting vertical super-amphiphobic surfaces, and air-in-hexadecane HDs impacting solid surfaces under-liquid (Supplementary Movie 2).

### Suppressed rebound of hollow droplets

The release height of the droplets was increased to characterize the bouncing dynamics at higher Weber numbers, $We = \rho V_0^2 R_0/\gamma$, where $\rho$ represents the density of the liquid (see in "Methods−Definition of Weber number" for discussion). Upon increasing the kinetic energy of impact, both SDs and HDs were observed to rebound (Supplementary Fig. 4 and Supplementary Movie 3). We compared the contact times ($t_c$) of SDs and HDs and made several notable observations. Firstly, $t_c$ was found to be independent of the impacting velocity ($V_0$) for both SDs and HDs (Supplementary Fig. 5a). Secondly, $t_c$ exhibited an increase with the characteristic radius $R_h$ in the form of $t_c \sim R_h^{3/2}$ (Supplementary Fig. 5b). As such, for both bouncing SDs and HDs, the contact time scales according to the inertial-capillary timescale, $\tau \sim (m/\gamma)^{1/2} \sim (\rho R_h^3/\gamma)^{1/2}$ with $m \sim \rho R_h^3$ (Supplementary Fig. 5c), consistent with previous studies[2,19]. Astonishingly, bouncing HDs displayed a shorter contact time compared to SDs by approximately 25% (Supplementary Fig. 5).

Despite the shorter contact time, we demonstrate that the rebound of HDs is still suppressed in comparison to SDs, as evidenced by their distinct dynamic behavior during impact. With the internal bubble, the HD exhibits a flying-saucer shape at its maximal deformation (Fig. 2a), instead of the pancake shape commonly observed for the SD. After the bounce, the HD is considerably less vibrant than the SD. The dynamics of the HD differ from those of the SD in three aspects: (i) The spreading factor $\beta = R_{max}/R_0$ (where $R_{max}$ is the maximum spreading radius, Fig. 2a) is smaller and decreases with $\Phi$ (Supplementary Fig. 6a), (ii) the retraction velocity $V_{ret}$ (defined as the time derivative of the contact radius $R_{contact}$, as shown in Fig. 2a) is higher (Fig. 2b) and increases with $\Phi$ (Supplementary Fig. 6b), and (iii) the restitution coefficient $\varepsilon = V_{reb}/V_0$ (where $V_{reb}$ is the rebound velocity) is lower and decreases with $\Phi$ (Supplementary Fig. 6c). These results suggest that the rebound is markedly suppressed by encapsulating a bubble in droplets. The reduced contact time of bouncing HDs is thus attributed to the smaller spreading factor and faster retraction velocity.

To clarify the influence of the internal bubble on the droplet dynamics, we performed a scaling analysis considering $\Phi$ and $We$. We assumed that the bubble consists of a hemisphere with a maximum radius of $R_{b-max}$ and a deformed spherical cap with the same maximum radius and a height of $h_0$ (Fig. 2a). To conserve the volume of the bubble, we had the relationship $R_{b-max}^3 + R_{b-max}^2 h_0 \sim R_b^3$, which leads to $(R_{b-max}/R_b)^2 \sim R_b/(R_{b-max} + h_0)$. In the case of a slightly deformed bubble, we can approximate $R_{b-max} \approx R_b$, giving us $R_{b-max} \sim R_b [R_b/(R_b + h_0)]^{1/2}$. The height $h_0$ can be determined as $h_0 \sim R_0 We^{-1/2}$ by balancing capillarity and gravity[18], $\gamma/h_0 \sim \rho a h_0$, with a reinforced gravitation acceleration of $a = V_0^2/R_0$. Substituting $R_b$ with $R_0 \Phi^{1/3}$, we can estimate $R_{b-max} \sim R_0 \Phi^{1/3}(1 + \Phi^{-1/3} We^{-1/2})^{-1/2}$. To model the maximally deformed HD composed of a hemispherical cap (with radius $R_{b-max}$) and a pancake-like lamella (with radius $R_{max}$ and thickness $h_0$), volume conservation of the HD yields $R_{b-max}^3 + R_{max}^2 h_0 \sim R_0^3$ (Fig. 2a). Using the estimated values of $R_{b-max}$ and $h_0$ from the previous analysis, we find that $R_{max}$ scales as $R_{max} \sim R_0 We^{1/4}[1 - \Phi(1 + \Phi^{-1/3} We^{-1/2})^{-3/2}]^{1/2}$. The spreading factor can then be determined based on these estimations:

$$\beta = C_1 We^{\frac{1}{4}}\left[1 - \Phi\left(1 + \Phi^{-\frac{1}{3}} We^{-\frac{1}{2}}\right)^{-\frac{3}{2}}\right]^{\frac{1}{2}}. \tag{1}$$

where $C_1$ is a pre-factor. When $\Phi = 0$, Eq. (1) reduces to $\beta \sim We^{1/4}$, identical to a previous prediction of $\beta$ for SDs[18,44]. Overall, our model provides a unified description of how $\beta$ increases with $We$ and decreases with $\Phi$, consistent with the experimental results (Supplementary Fig. 6a). Figure 2c shows the experimental results against the

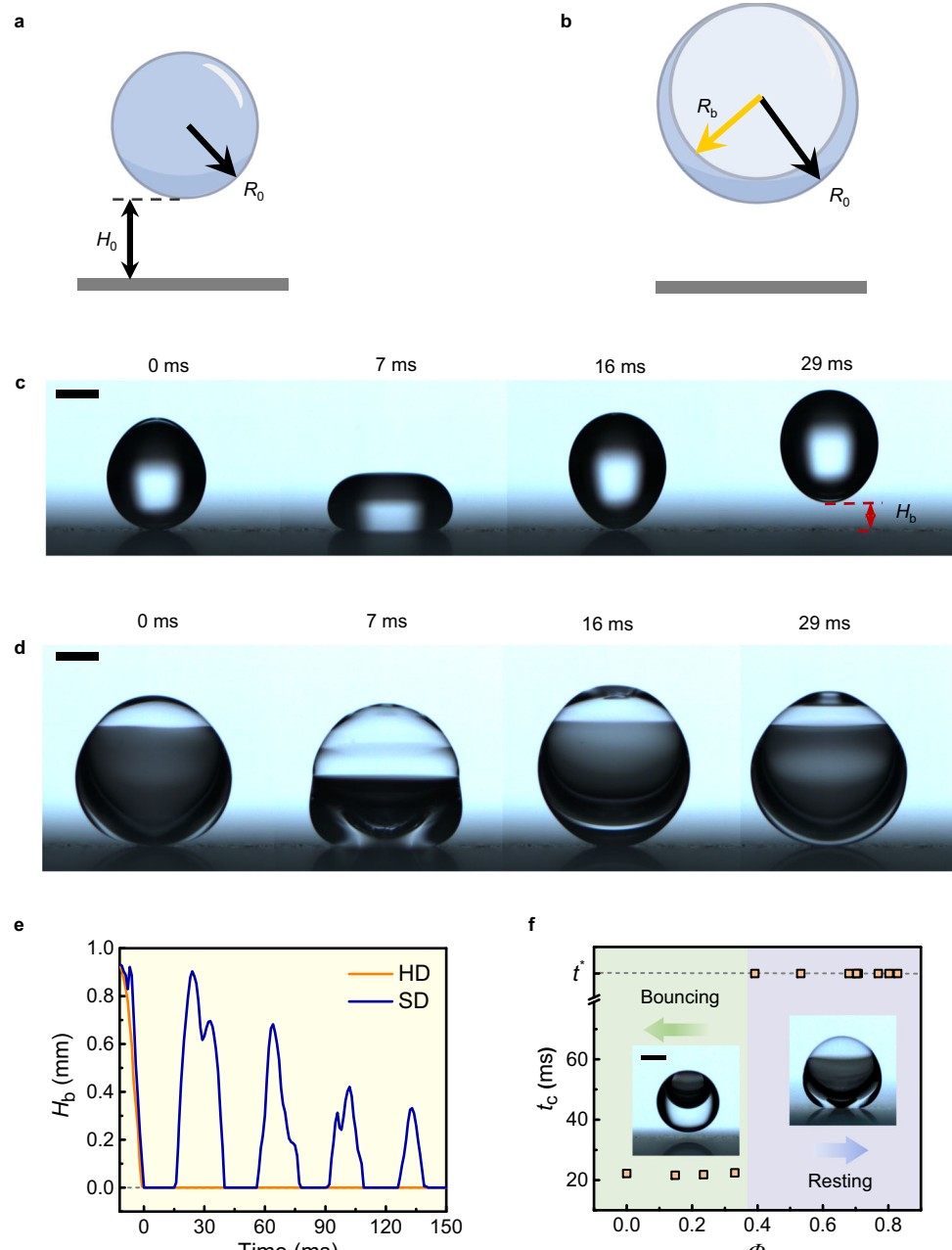

**Fig. 1 | Non-rebound of an impinging hollow droplet. a, b** Schematic of a single-phase droplet (SD) of radius $R_0$ (**a**) and a hollow droplet of apparent radius $R_0$ and bubble radius $R_b$ (**b**), released from the initial height $H_0$. **c, d** Snapshots of the SD (**c**) and HD (**d**) impacting a super-repellent surface released from a height of $H_0 = 0.9$ mm. The SD takes off, whereas the HD rests on the surface after impact. **e** The plot of the droplet's bottom height ($H_b$) versus time for the two events in (**c**)

and (**d**). **f** Dependence of the contact time $t_c$ on bubble volume fraction $\Phi$. $\Phi = 0$ stands for SD while $\Phi > 0$ for HD. In the case of the non-rebound HD, the contact time is set to be equal to the droplet's complete evaporation time $t^*$, which was approximately 7200 s in experiments. Insets, snapshots of a bouncing HD with $\Phi = 0.2$ (left) and a resting HD with $\Phi = 0.8$. The liquid surface tension is 53 mN m$^{-1}$ in (**c**)–(**f**). Scale bars, 1 mm.

theoretical prediction of $\beta$. The comparison validates Eq. (1), and the linear fitting indicates that $C_1 = 0.73$.

After maximal spreading, the three-phase contact line of the HD retracts at a constant velocity $V_{ret}$ (Fig. 2b) that is determined by the balance between the capillary and inertial forces, $\gamma/h_0 \sim \rho V_{ret}^2$. Considering that both the HD and SD have the same mass and volume of the liquid phase, the lamellar thickness ($h_0$) of the HD is smaller compared to that of the SD. This is due to the liquid being pushed away by the central bubble, resulting in the spread of the liquid into a lamella with a larger maximum radius (see Fig. 2a). To obtain a more accurate estimation of $h_0$ that highlights the differences between HD and SD by taking the bubble into consideration, we need to re-evaluate the

determination of $h_0$ by considering the force balance of $\gamma/h_0 \sim \rho a h_0$. The reinforced gravitation acceleration is reconsidered as $a \sim V_0^2/(R_0 - R_b) = V_0^2/R_0(1 - \Phi^{1/3})$. Consequently, $h_0$ is updated into $h_0 \sim R_0 We^{-1/2}(1 - \Phi^{1/3})^{1/2}$, which is a function of and decreases with both $We$ and $\Phi$. This validates the assertion that the lamellar thickness of the HD is smaller than that of the SD. As a result, the driving capillary force ($-\gamma/h_0$) is larger, leading to an increased retraction velocity ($V_{ret}$) for the HD. Using the value of $h_0$ estimated above, we determine $V_{ret}$ to be,

$$V_{ret} = C_2 \left(\frac{\gamma}{\rho R_0}\right)^{\frac{1}{2}} We^{\frac{1}{4}} \left[1 - \Phi^{\frac{1}{3}}\right]^{-\frac{1}{4}}. \tag{2}$$

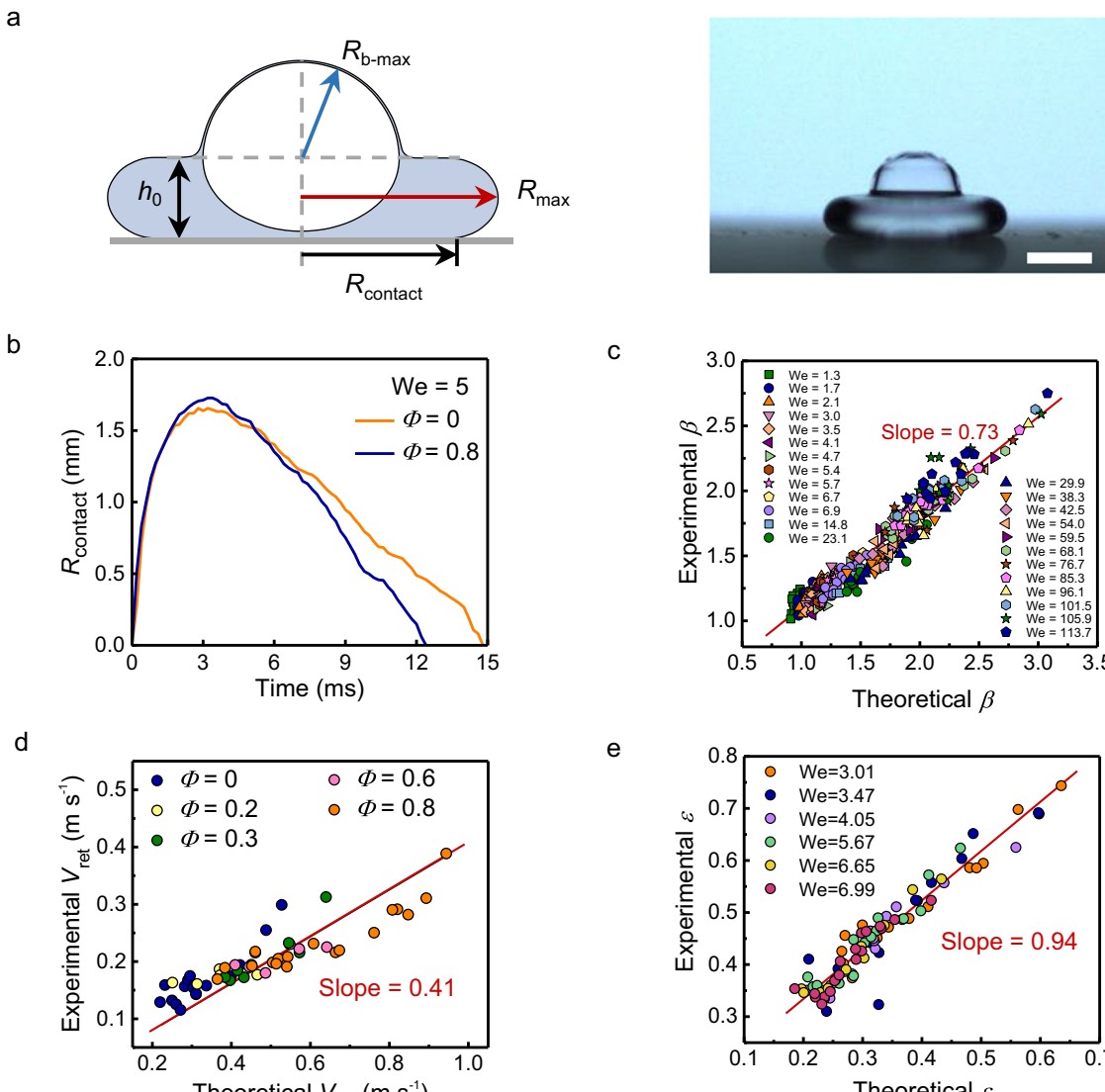

**Fig. 2 | Bouncing dynamics of droplets. a** Schematic (left) and picture (right) showing the HD spreading to its maximum radius $R_{max}$, with the lamella height $h_0$, bubble radius $R_{b\text{-}max}$, and contact radius $R_{contact}$. **b** The plot of $R_{contact}$ as a function of time when the SD and HD impact a superhydrophobic surface. The retraction velocity of the HD is higher than that of the SD during droplet receding. **c**–**e** Comparison of the theoretical prediction with the experimental value of the spreading factor $\beta$ (**c**), retraction velocity $V_{ret}$ (**d**), and restitution coefficient $\varepsilon$ (**e**) for HD. In (**b**)–(**e**), the data was measured from experiments of droplets impacting superhydrophobic surfaces when $We < 10$ and superamphiphobic surfaces when $We > 10$, where the liquid surface tension was 53 and 35 mN m$^{-1}$, respectively. Scale bar, 1 mm.

The $V_{ret}$ estimated by the proposed model is consistent with the experimental results shown in Fig. 2d, which yields a pre-factor of $C_2 = 0.41$. For SDs with $\Phi = 0$, Eq. (2) reduces to $V_{ret} \sim (\gamma/\rho R_0)^{1/2} We^{1/4}$, consistent with the previous studies[5,44].

The characteristic take-off velocity of a droplet scales as[18] $V_c \sim (\gamma/\rho R_0)^{1/2}$. However, the bubble acts as a baffle that obstructs the upward liquid flow in the HD. Assuming that the bubble-induced resisting momentum scaling as $(R_{b\text{-}max}/R_{max})^{1/2}mV_c$, the momentum balance can be obtained as $mV_{reb} = (1 - R_{b\text{-}max}^2/R_{max}^2)mV_c$ for the HD. By substituting the estimated values of $R_{b\text{-}max}$ and $R_{max}$ into the analysis, we can derive the rebound velocity $V_{reb}$ as a function of $We$ and $\Phi$, which is $V_{reb} \sim [1 - We^{-1/2}\Phi^{2/3}(A - \Phi A^{-1/2})^{-1}]V_c$ with $A = 1 + \Phi^{-1/3}We^{-1/2}$. By defining $\varepsilon = V_{reb}/V_0$, the restitution coefficient can be predicted as,

$$\varepsilon = C_3 We^{-\frac{1}{2}}\left[1 - We^{-\frac{1}{2}}\Phi^{\frac{2}{3}}\left(A - \Phi A^{-\frac{1}{2}}\right)^{-1}\right]. \quad (3)$$

Equation (3) suggests that $\varepsilon$ decreases with both $We$ and $\Phi$, consistent with the experimental results (Supplementary Fig. 6c). When $\Phi = 0$, Eq. (3) reduces to $\varepsilon \sim We^{-1/2}$, which represents a conventional scaling for the SD. Figure 2e shows that the results obtained experimentally and theoretically agree with each other, with a pre-factor of $C_3 = 0.94$.

**Mechanism for rebound suppression**

Subsequently, we discuss the mechanism underlying rebound suppression. A superficial conjecture may attribute the suppressed rebound to the energy dissipation ($E_d$) in compound fluid systems[45,46]. Numerical simulations have shown that the energy dissipation ratio of HDs is even smaller than that of SDs during impact. Supplementary Fig. 7a, b demonstrate that only approximately 4% of the total energy ($E_0$) is lost during the spreading and recoiling of a rebounded HD at an impact velocity of $V_0 = 0.270$ m s$^{-1}$, while this energy loss proportion is reduced to around 1% when the HD does not rebound at $V_0 = 0.135$ m s$^{-1}$. It is noteworthy that the SD bounces despite having a

higher energy dissipation ratio compared to the HD at both impact velocities. Thus, the suppression of rebound cannot be attributed to energy dissipation.

With negligible energy dissipation, the sum of kinetic energy ($E_k$), surface energy ($E_s$), and potential energy ($E_p$) remains nearly conserved, and they can be transformed among one another (Fig. 3). However, HDs and SDs follow different energy conversion pathways. For bouncing SDs, almost all the kinetic energy is converted to surface energy during spreading (Fig. 3a). In contrast, the non-rebound HDs maintain a high and nearly constant value of kinetic energy after impact (Fig. 3b). This disparity is also evident at higher impact velocities, where both HDs and SDs rebound (Fig. 3c, d). Numerical simulations reveal that only approximately 30% of the kinetic energy is converted to surface energy during the spreading stage of bouncing HDs (Fig. 3d). The suppression of HD rebound with such a low conversion efficiency from kinetic energy to surface energy calls for further exploration.

The rebound suppression can potentially be understood by the distribution of the flow field (Fig. 4 and Supplementary Movie 4). In the SD, the liquid presents a unidirectional, upward internal flow responsible for the take-off. In contrast, in the HD, the distribution of the flow velocity is inhomogeneous during droplet retraction. The encapsulated bubble always hovers in the upper part of the HD owing to buoyancy. The retraction-induced upward flow is blocked by the bubble, and the direction of the flow velocity may change from upward to downward. As both the inner and outer gas-liquid surfaces deform, the direction of the flow velocity alternately shifts between upward and downward. The velocity oscillations experienced by HDs significantly attenuate the net momentum, resulting in either a negative net momentum (Supplementary Fig. 7c) for non-rebound or a low positive value (Supplementary Fig. 7d) that lifts

the HD with a suppressed height. This scenario is distinct from that of SDs, where higher rebound is associated with larger net momentum (Fig. 4).

Two capillary effects compete during the HD retraction. The outer-droplet surface retracts to push the liquid upward, whereas the dimple-shaped bottom of the inner-bubble surface produces a Laplace pressure ($\Delta P$) that prevents the liquid from upwelling (Fig. 5a). The dimple-shaped surface is reminiscent of the shape of counter-jet reported in previous studies[21,30–32]. However, in our case, the upward growth of the dimple is inhibited due to the low *We*, preventing the bubble from breaking up (Figs. 4 and 5a). Considering these aspects, the recoiling HD can be modeled as a double-spring system in which a mass (representing the liquid) is sandwiched between a lower spring (representing the outer-droplet surface) with stiffness $k$ and length $l_o$ and an upper spring (representing the inner-bubble surface) with stiffness $k$ and length $l_i$ (Fig. 5b). The two springs counteract each other: the elastic energy released by one spring is partly absorbed by the other. The evolutions of the inner surface area ($S_i$) and outer surface area ($S_o$) exhibit opposite trends when the HD retracts (Fig. 5c, d), which validates the results of the previous analysis. The decrease in $S_o$ (or $S_i$) is compensated by the increase in $S_i$ (or $S_o$), which accounts for the increase in the total surface area ($S_i + S_o$) during HD retraction, in contrast to the SD case (Supplementary Fig. 7e, f).

The double-spring model is characterized by a reduced effective elasticity. Unlike in the case of SDs, where released surface energy is directly converted into kinetic energy, impacting HDs exhibit an extra energy conversion between the inner and outer surfaces. In this process, the surface energy released from one surface is absorbed by the other, resulting in a reduced conversion efficiency between surface energy and kinetic energy. With the two surfaces analogous to the two

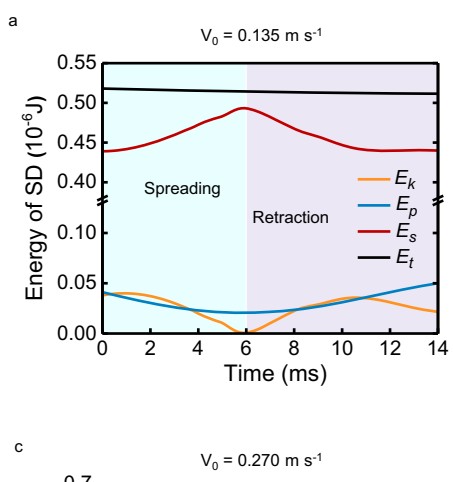

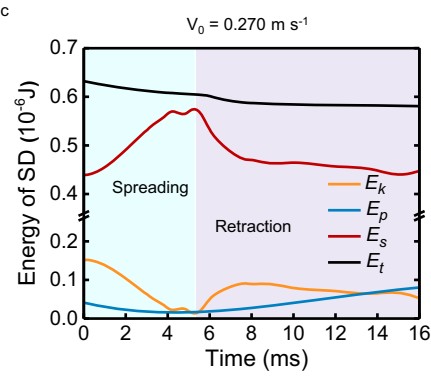

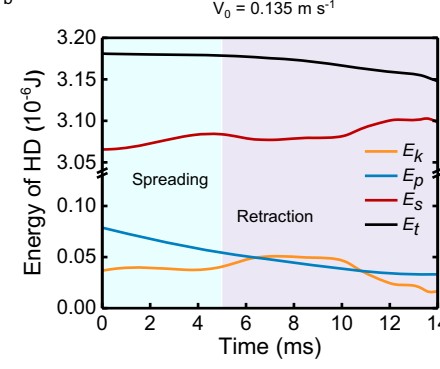

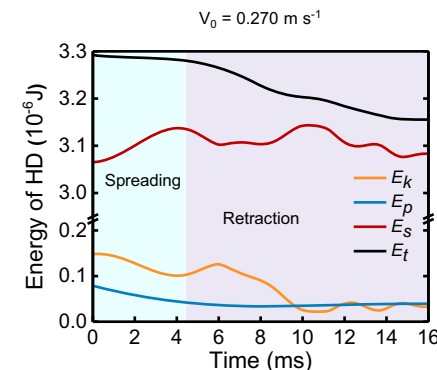

**Fig. 3 | Energy changes during droplet impact by numerical simulation.**
**a, b** Energy changes of the SD (**a**) and HD (**b**) at an impact velocity of $V_0 = 0.135\,\text{m s}^{-1}$. The SD bounces while HD rests on the surface after impact. $E_k$, $E_p$, $E_s$, $E_t$ represent kinetic energy, potential energy, surface energy, and their sum, respectively. $E_t = E_k + E_p + E_s$. **c, d** Energy changes of the SD (**c**) and HD (**d**) at a higher impact velocity of $V_0 = 0.270\,\text{m s}^{-1}$. Both SD and HD bounce off the surface after impact.

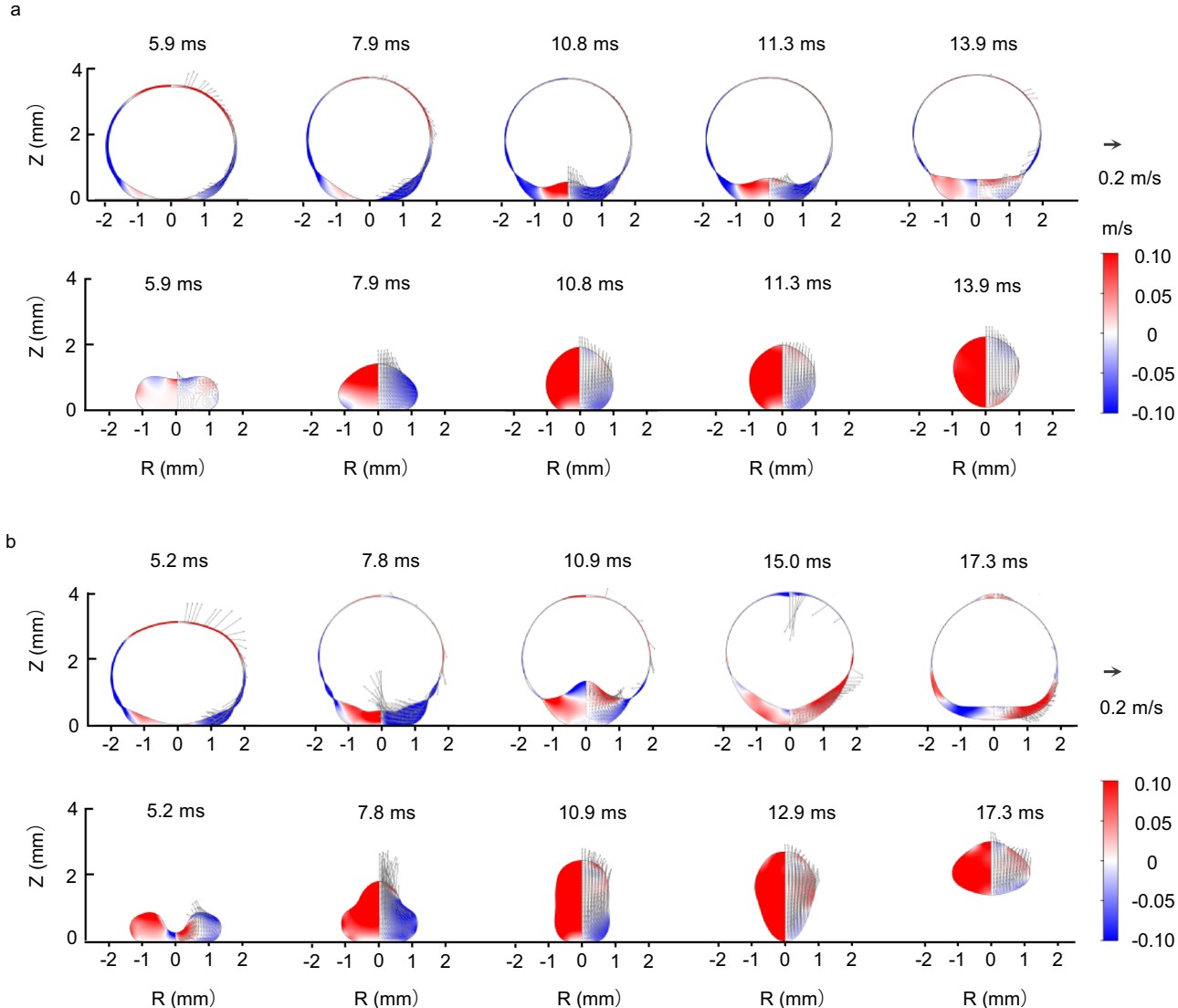

**Fig. 4 | The internal flow of impacting droplets. a, b** The internal flow of HD (upper row) and SD (lower row) during droplet retraction at an impact velocity of 0.135 m s$^{-1}$ (**a**) and 0.270 m s$^{-1}$ (**b**) by numerical simulation. Arrows indicate velocity vectors. The colored left and right halves of each snapshot show the vertical and radial components of the fluid velocity, respectively. Red represents the positive velocity and blue for the negative velocity. In both cases, the SD exhibits bouncing, while the HD only demonstrates bouncing at a higher impact velocity of 0.270 m s$^{-1}$. The HD's bouncing height is significantly lower than that of the SD.

competing springs in the double-spring system, we can assume that $k \sim \gamma$, $l_o \sim R_O$, and $l_i \sim R_b$. We propose a qualitative estimation of the apparent surface tension ($\gamma_a$), which accounts for the overall counteractive capillary effect as the difference in capillarity between the two surfaces, $R_O\gamma_a \sim (R_O\gamma - R_b\gamma)$. Scaling analysis suggests that $\gamma_a \sim (R_O - R_b)\gamma/R_O = (1 - \Phi^{1/3})\gamma$. This apparent surface tension is lower than the actual liquid surface tension ($\gamma$) and decreases with the bubble volume fraction ($\Phi$).

Based on this reasoning, the apparent surface tension decreases asymptotically to zero when the bubble volume fraction approaches unity. This result suggests that high-$\Phi$ HDs can behave as droplets with nearly zero surface tension when impacting super-repellent surfaces. The rebound suppression can be strengthened by increasing the number of bubbles in the HD (Supplementary Movie 5), in which case the two-spring model extends to a multiple-spring model (Supplementary Fig. 8) with enhanced counteractive capillary effects.

## Phase diagram

According to the energy-conversion argument, the effects of the inner-bubble surface are twofold: (i) The dimple-like surface deformation absorbs part of the kinetic energy, which attenuates the upward motion of the liquid, and (ii) the recovery of the inner-surface deformation converts the surface energy back into the kinetic energy of the liquid flowing downward. Despite the high total kinetic energy of the HD, only a small fraction is responsible for droplet rebound. We thus speculate that droplet rebound will be completely suppressed when a critical amount of the impacting kinetic energy ($\sim R_h^3\rho V^2$) is absorbed by the inner-bubble surface ($\sim R_b^2\gamma$). This hypothesis implies an energy-balance scaling of $(R_O^3 - R_b^3)\rho V^2 \sim R_b^2\gamma$. Therefore, a critical Weber number pertaining to the transition between the rebound and non-rebound conditions can be found:

$$We = C_4\Phi^{\frac{2}{3}}/(1 - \Phi). \tag{4}$$

with $C_4 \approx 1/7$ determined empirically from the experimental results. The phase diagram in Fig. 6 shows that our model can accurately predict the bouncing dynamics of HDs.

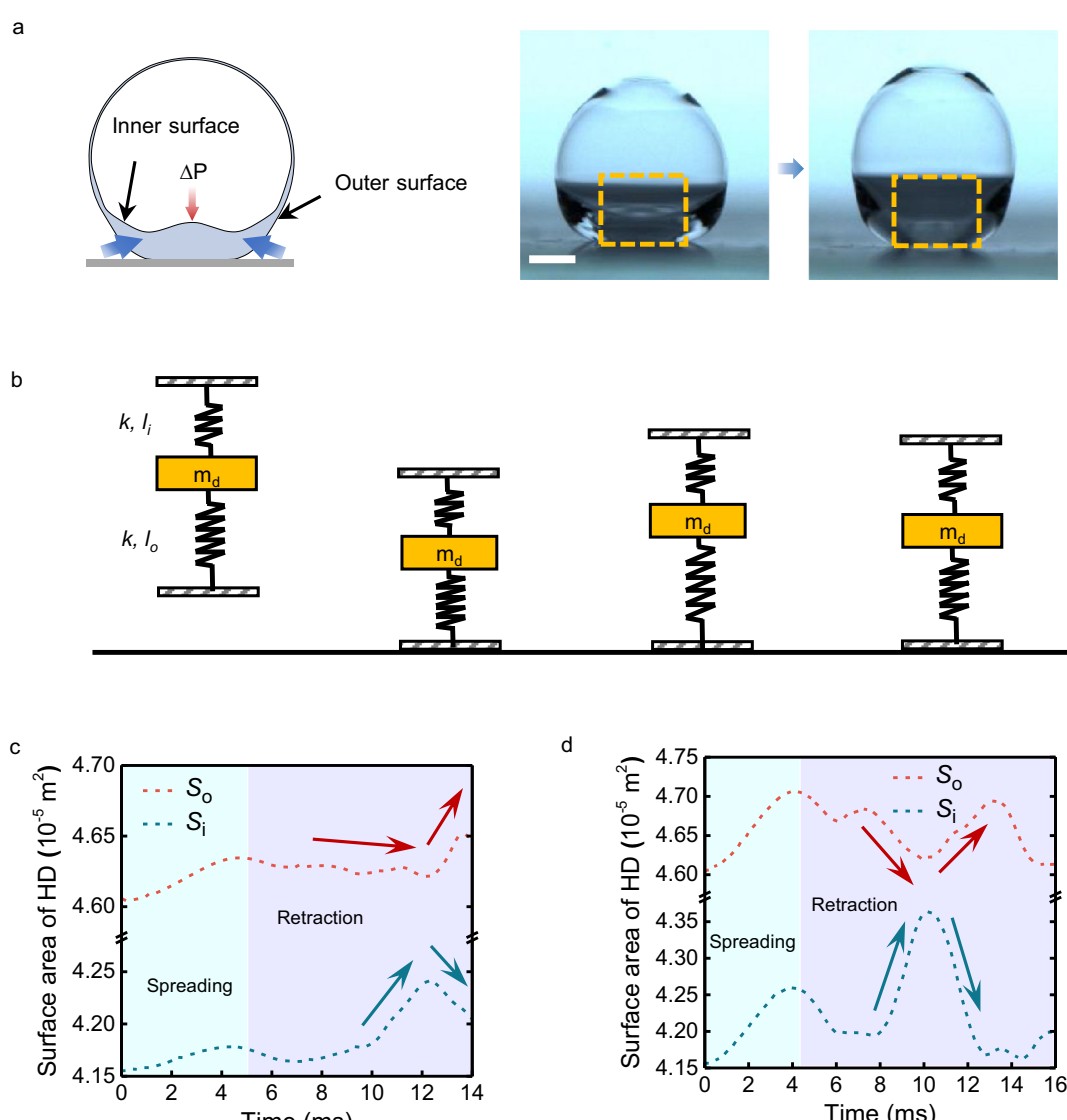

**Fig. 5 | Counteractive capillary effects for rebound suppression. a** Schematic (left) and pictures displaying the asynchronized deformation of the inner and outer surfaces, which induces capillary flows opposing each other. The dashed yellow boxes highlight the dynamic changes of the dimple shape on the inner surface.

**b** Schematic of the double-spring system that depicts the counteractive capillary effects in HDs. **c**, **d** Temporal evolution of the inner ($S_i$) and outer ($S_o$) surface areas for HDs at the impact velocity of 0.135 m s$^{-1}$ (**c**) and 0.270 m s$^{-1}$ (**d**) by numerical simulation. Scale bar, 1 mm.

## Discussion

Overall, the results of this study highlight that the droplet structure can be simply modified to suppress the rebound, without altering the droplet composition or surface properties. Notably, bubbles are spontaneously entrained during droplet impact[3]. The influence of these entrained bubbles on the bouncing dynamics may bring awareness of their new scientific interest. The enhanced deposition of hollow droplets on super-repellent surfaces can potentially be leveraged, for example, for increasing the self-cleaning efficiency via the enlarged contact area of the sliding/rolling droplets (Supplementary Fig. 9 and Supplementary Movie 6) and increasing the efficiency of spray cooling owing to the prolonged contact time. Similar phenomena can also be observed in macroscopic systems. For example, the rebound of a compound balloon (containing water and an air-filled balloon) is conspicuously suppressed (Supplementary Fig. 10 and Supplementary Movie 7) compared with that of a single-phase balloon (filled with only water). This finding can extend the applications of shock-absorbing hollow systems to a wider horizon, for instance, to develop throw-type

fire-extinguishing balls[43] that must be precisely positioned at the target fire to release the interior content without bouncing.

## Methods

### Hollow droplet generation

A co-flow microfluidic device was used to generate hollow droplets, in which air and liquid were used as the inner and outer phase fluids, respectively (Supplementary Fig. 1). The inner diameter of the inner capillary ranged from 0.06 mm to 0.1 mm, and that of the outer capillary ranged from 0.25 mm and 0.9 mm. Two syringe pumps (Longer Pump, LSP01-1A) were used to drive the two working fluids and control the flow rates. To maintain the dripping mode for the generation of monodisperse hollow droplets, the flow rate of the outer phase liquid was set in a range between 2 mL h$^{-1}$ and 8 mL h$^{-1}$. The volume fraction of the encapsulated bubble was adjusted by controlling the inner airflow in a range between 1 mL h$^{-1}$ and 5 mL h$^{-1}$.

Water, with sodium dodecyl benzene sulfonate (SDBS, Macklin) as the surfactant, was used as the outer fluid for hollow droplet impact on superhydrophobic, superamphiphobic, and Leidenfrost-state

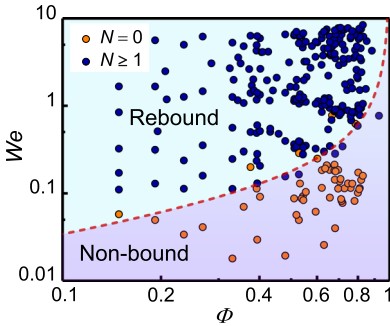

**Fig. 6 | Phase diagram.** Two regimes, rebound and non-bound, are observed and tunable by the Weber number $We$ and bubble volume fraction $\Phi$. The dashed line presents the theoretical prediction of the boundary between the two regimes, as calculated using Eq. (4) with a fitting prefactor of $C_4 = 1/7$. $N$ indicates the number of droplet rebounds.

surfaces. In addition to pure water, aqueous solutions with three different SDBS concentrations were used in experiments: impingement of 0.003 wt% SDBS (surface tension of 53 mN m$^{-1}$) droplets on superhydrophobic surfaces; impingement of 0.1 wt% SDBS (surface tension of 43 mN m$^{-1}$) droplets on smooth aluminum plates in the Leidenfrost regime; impingement of 0.2 wt% SDBS (surface tension of 35 mN m$^{-1}$) droplets on superamphiphobic surfaces. Besides, $n$-hexadecane (98%, Aladdin) was used as the outer liquid phase to produce air-in-oil HDs for impact experiments. The selection of various liquids and solid surfaces allowed us to demonstrate the universality of the phenomenon of rebound suppression by HDs across different liquid compositions and solid surface properties.

### Fabrication of superhydrophobic surfaces

The superhydrophobic coating suspension was made by dispersing fumed silica nanoparticles (Aerosil R805, Evonik) in ethanol at a concentration of 2 wt%. A glass substrate was cleaned with deionized water first and then ethanol in an ultrasonic bath. After drying, the substrate was coated with the superhydrophobic solution twice by dip-coating. Then the coated surface was baked at 60 °C for 10 min in a drying oven.

### Fabrication of superamphiphobic surfaces

The superamphiphobic coating was fabricated by the following steps. First, 15 mL ammonia solution (28–30%, Aladdin) was added to 45 mL ethanol (absolute, Anaqua), which was stirred at an agitation speed of 400 RPM and heated to a temperature of 60°C in a water bath. Then, 2 mL tetraethyl orthosilicate (>99%, Macklin) was added into the solution drop by drop. When the solution became opaque, 0.1 mL 1H,1H,2H,2H-perfluorodecyltriethoxysilane (>98.0%, TCI) was added to the mixture. After 24 h of reaction, a superamphiphobic solution was obtained, which contained fluorinated silica nanoparticles. A glass substrate was dip-coated with the solution and then incubated at 80 °C for 1 h in a drying oven. The coating and drying processes were repeated two to three times until a uniform superamphiphobic nanoparticle coating was obtained on the glass surface.

### Droplet impact

Hollow droplets impact the solid super-repellent surface directly after their generation. The height of the droplet impact was tunable by a sliding rail. The impact process was recorded using a high-speed camera (Fastcam Mini, Photron) at a frame rate ranging from 5000 frames per second (fps) to 6520 fps. The shape variation of droplets during impingement was analyzed using ImageJ (National Institutes of Health).

### Leidenfrost experiment

A smooth, flat, and well-polished aluminum plate was placed on a copper heater equipped with a temperature controller. The real-time surface temperature of the aluminum plate was measured using a $K$-type thermocouple and monitored using a recorder. In Leidenfrost experiments, the temperature of the aluminum plate was controlled at 210 ± 5 °C. A tilting stage was used to maintain the tilt angle at 10° for droplets impacting inclined surfaces.

### Self-cleaning

A superamphiphobic surface was covered with a layer of Fe$_3$O$_4$ nanoparticles (Zhonghangzhongmai Metal Material Co. Ltd) as the dust. The surface was tilted at an angle of 10° by a tilting stage. A hollow droplet impacted the tilted surface and absorbed the dust particles when rolling down the surface.

### Balloon impact

The compound core-shell balloon was made by containing water as the shell and a hollow balloon (filled with air) as the core. In the control group, a single-phase balloon was filled with only water. For a fair comparison, the two balloon systems contained the same mass of water (500 g) and were released from the same height to impact the ground.

### Definition of Weber number

In the study, the Weber number ($We$) was defined using the density of the liquid phase ($\rho$) rather than the apparent density ($\rho_a$) of the HD. The mass of the HD was calculated by $m = 4\pi R_0^3 \rho_a / 3 = 4\pi R_h^3 \rho / 3$ where $R_h^3 = R_0^3 - R_b^3$. We then have $\rho_a = R_h^3 \rho / R_0^3 = (R_0^3 - R_b^3)\rho / R_0^3 = (1 - R_b^3 / R_0^3)\rho = (1 - \Phi)\rho$. Based on this, the $\rho_a$-based Weber number is defined as $We_a = \rho_a V_0^2 R_0 / \gamma = We(1 - \Phi)$ where $We = \rho V_0^2 R_0 / \gamma$. It is important to note that $We_a$ depends on $\Phi$ while $We$ does not. To investigate the bouncing behavior of HDs systematically and explicitly, it is crucial to consider independent groups of non-dimensional parameters (such as $We$ and $\Phi$). With this consideration, the $\rho$-based $We$ was used instead of $\rho_a$-based $We_a$ in this study, and this choice does not affect the accuracy of the models presented in Eqs. (1)–(4), because $We_a$ can be readily determined using the values of $We$ and $\Phi$.

### Numerical simulation

For the numerical simulation of the droplet impact, we used the open-source code Basilisk (basilisk.fr)[47]. The code uses a second-order in space, time-splitting projection method for incompressible two-phase flows, and a geometric Volume of Fluid (VoF) method to accurately track the liquid-gas interface.

Basilisk discretizes the domain using an Octree grid (Quadtree grid in 2D), which can dynamically refine and coarsen its resolution during computation. This adaptive mesh refinement (AMR) feature allows for the finest grid resolution around the droplet interface while keeping relatively lower resolutions elsewhere to reduce the computational cost.

In our case, the simulation was enforced to be axis symmetry, so simulating a 2D cross-section of the droplet is adequate. Given a square numerical domain of the side length $L$, a cell refined by $n$ times has the side-length of $L/2^n$. With $L = 5.74$ mm and the maximum level of the refinement $n = 13$, the solver can resolve spatial scales as small as 0.7 microns. This fine spatial resolution is crucial to avoid numerical breakups, and Basilisk's AMR capability makes it possible at a reasonable computational cost.

## Data availability

The data that support the findings of this study are available from the corresponding authors. Source data are provided with this paper.

## Code availability

The code that supports the findings of this study is available from the corresponding authors.

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

## Acknowledgements

The financial support from the Research Grants Council of Hong Kong (Project No. CityU21213621), Shenzhen Science and Technology Program (Project No. JCYJ20220530140812028), and City University of Hong Kong (Project Nos. 9610502 and 7005936) is gratefully acknowledged.

## Author contributions

P.Z. conceived the research. P.Z. and Y.Z. designed the experiments. W.Z. fabricated the superamphiphobic coating. S.W. and Y.Z. fabricated the microfluidic device for droplet generation. Y.Z. performed the experiments. P.Z. performed the theoretical analysis. C.Z. performed the numerical simulation. P.Z., Y.Z., and C.Z. analyzed the data. P.Z. and Y.Z. wrote the manuscript. P.Z. supervised the research. All authors commented on the paper.

## Competing interests

The authors declare no competing interests.
