## [Peer Review File · Nature Communications]

Review on “Non-rebound of hollow droplets on super-repellent surfaces”

In this article, Zhou *et al.* reports an intriguing idea for inhibiting drops from bouncing off very liquid-repellent substrates. By including a large air bubble inside a water droplet, its jumping can be strongly suppressed on surfaces that are well-known for their repellent properties. I find this work nicely thought provoking as we often try to make superhydrophobic surfaces as repellent as possible. For example, the scientific discussion is often on how to further *reduce* the contact time between a bouncing drop and the substrate. Here, Zhou *et al.* argue for the need of a non-rebound (i.e., “infinite” contact time) yet highly liquid-repellent (i.e., low friction) substrate, since this will improve, e.g., the self-cleaning properties when keeping the impinging water drops on the sample while simultaneously allowing them to easily slide off. The inclusion of an air drop in the water drop is a beautifully simple solution to this problem.

In my opinion, the paper is very well written, interesting, and easy to read. The experiments have been carefully conducted with beautiful high-speed movies and the authors make a strong effort in discussing the possible reasons to why hollow drops do not bounce. Below, I list minor comments on how to further improve this work. After these changes, I warmly recommend this paper for publication in Nat. Commun. I believe the aesthetics of the experiments combined with the quite general system of a drop bouncing (or not) off a surface will attract a broad audience. Furthermore, the findings are important for the fluid dynamics and surface science community working on superhydrophobic surfaces, wetting, and drop dynamics.

Minor comments

1. Regarding the surface tension.

- In the Methods section, you mention that the concentration of SDBS was “up to 0.2 wt%”. What does “up to” mean? Was the surface tension kept constant in all experiments or was there a big variation in the surface tension in different bouncing experiments? If the concentration is different in different experiments, the exact surface tension values should be mentioned in the captures of all figures and discussed in detail in the Methods section.
- Why do you show profiles of drops with *different* SDBS concentrations in Supp. Fig. 2b? Did a 0.2 wt% drop not bead up on a superhydrophobic surface?
- Was SDBS added also to the SD water drops in the bouncing experiments of the paper? Please specify this in the main text.
- Please mention the range of surface tensions used (35-?? mN/m) in the main text (Lines 55-56, p. 3) so that the reader understands that this is significantly below that of water.

2. I cannot easily follow how you reached the expression for $\gamma_e \sim (R_0 - R_b)\gamma/R_0$ on Line 157 (p. 7). Since this is an important result in describing why the effective surface tension can be considered much smaller in your system, its derivation would need a more extensive explanation.
3. I assume that the pre-factors C in Eqs. 2, 3 and 4 are not to be considered the same? If this is correct, I would suggest giving these constants different variable names (e.g., A, B, and C or C₁, C₂ and C₃) to avoid confusion.
4. Reg. Fig 4:
 - This is a beautiful graph! Can you please mention in the caption that the dashed line is a **fit** of the theoretical prediction of Eq. 4.
 - You write “N indicates the time of droplet rebound.”. Do you mean “N indicates the number of droplet rebounds.”?
5. In the discussion, you claim that you can “suppress the rebound, without altering the droplet composition or surface properties”. However, you do significantly change the surface tension of water by adding the stabilising surfactant (SDBS, see my comment in 1 above). I would recommend rephrasing this sentence.
6. I find the final comparison with compound balloons delightful!
7. A similar system of hollow droplets impacting a solid substrate has been studied before (see papers below). Although these papers report on the spreading dynamics and not rebound inhibition, I think a reference to their work would be appropriate.
 - “Hollow droplets impacting onto a solid surface”, I. P. Gulyaev & O. P. Solonenko, *Experiments in Fluids* **54**, 1432 (2013). <https://doi.org/10.1007/s00348-012-1432-z>
 - “Impact dynamics of air-in-liquid compound droplets”, D. P. Naidu and S. Dash, *Physics of Fluids* **34**, 073604 (2022), <https://doi.org/10.1063/5.0096599>.

Reviewer #2 (Remarks to the Author):

The manuscript concerns a study about droplets impacting on superhydrophobic surfaces. In particular, the inclusion of a bubble inside the droplet, as the authors claim, suppresses the rebound phenomenon observed when a droplet impacts on a superhydrophobic surface. The suppression of the rebound is interesting however the study seems premature, in the sense that the analysis is not clear and does not give a picture of what really happens.

For example, the inclusion of the bubbles notably changes the apparent density of the composite droplet. And when the Weber number is used to compare the single and the composite droplet, which density is used to calculate We ?

Also, in figure 1f the non-rebound is observed when the ratio, liquid/bubble volume, Φ , > 0.4 . At this magnitude Φ , the composite system is completely different than the one of the single liquid. I mean that the systems are not comparable. We could mostly term it bubble and not droplet with a small bubble inside it.

The analysis in page 6 is not convincing. The argument that although the energy dissipation is very small but the velocity distribution, in the composite system, is such that rebound is suppressed is not solid.

The importance of using a relatively big bubble inside a droplet suffers from the fact that what suppression is only observed when gravity is present, i.e. in free fall. What happens in the case of a impact in side wall. Then the position of this big bubble could unpredictably change.

Reviewer #3 (Remarks to the Author):

The article by Zhou et al. approaches the classical problem of bouncing of water drops on a water-repellent surface, but with a twist: the drop in this case has a bubble encapsulated within. The authors compare the bouncing dynamics in the two cases to conclude that bubble inclusions may be utilized to trigger suppression of bouncing, without enhancing dissipation. The authors map this effect over Weber numbers varying over three orders of magnitude and supplement their arguments further with numerical simulations.

The work is (mostly) adequate, so is the writing (I also appreciate the video of the balloon). However, the observations in this case are perhaps better suited in a droplet-specific journal. I would just add a couple of remarks, which the authors could consider for their future endeavors with this work.

First, I think it would be wise to plot the actual contact time values, which the authors do not plot anywhere. To compare it with the original plot of Richard et al. which reported the Hertzian nature of the bouncing dynamics for the first time, they could plot the contact time against V for different ϕ (the ratio of the bubble to total volume). If dissipation is indeed negligible, as the authors claim, then they should see flat horizontal lines for different ϕ and even recover the line of Richard et al. for $\phi = 0$. Similarly, the authors should plot the contact time against the radius on a log-log plot to check if the scaling is indeed $3/2$. Finally, for the case presented by the authors, as we increase f for a given R_0 , we decrease the volume of water, yet increase the contact time. This is in sharp contrast to droplets impinging on superhydrophobic macrottextures (like lines or points), where the volume is diminished too (via reorganization) leading to a decrease in contact time (Bird et al. Nature 2013 and Gauthier et al. Nat. Commun. 2015). This needs to be discussed – why decreasing volume does not decrease the contact time for the author’s case, as one would expect from Quéré’s inertia-capillary scaling (Richard et al. Nature 2002). The increase in the contact time could be quantified as a function of ϕ to elaborate on this.

Reviewer #4 (Remarks to the Author):

The paper presents an experimental study on drop impact on phobic surfaces, showing how the presence of bubbles can promote rebound suppression.

Although the effect is limited to a relatively narrow region of We numbers (see Fig. 4), with the non-rebound to rebound threshold changing from ~ 0.1 to ~ 1 , the effect appear to be consistent and observed also in other conditions, such as Leidenfrost boiling, as well as other macroscopic systems, such as water balloons.

I also appreciate that the authors have included a simple spring model, based on a two-spring system, that nonetheless seems to reproduce well the experimental data.

I have no comments on the paper, and I recommend its publication.

Reviewer #5 (Remarks to the Author):

Non-rebound of hollow droplets on super-repellent surfaces

The manuscript reports differences in the impact behavior of a simple and an air-encapsulated hollow droplet on superhydrophobic substrates.

There have been several papers over the last 2-3 years that have reported experimental observations and numerical analysis of impact of hollow droplets on different substrates

1. Nasiri, M., Amini, G., Moreau, C. and Dolatabadi, A., 2023. Flattening of a hollow droplet impacting a solid surface. *Journal of Fluid Mechanics*, 962, p.A1.
2. Nasiri, M., Amini, G., Moreau, C. and Dolatabadi, A., 2021. Hollow droplet impact on a solid surface. *International Journal of Multiphase Flow*, 143, p.103740
3. Naidu, D.P. and Dash, S., 2022. Impact dynamics of air-in-liquid compound droplets. *Physics of Fluids*, 34(7), p.073604.
4. Wei, Y. and Thoraval, M.J., 2021. Maximum spreading of an impacting air-in-liquid compound drop. *Physics of Fluids*, 33(6), p.061703.

Please note that this is not the comprehensive list of papers and there are several more. There are no citations to these papers in the manuscript. I am not sure if the authors are aware of these works.

- During impact of a hollow droplet on a substrate, a counterjet forms in addition to the liquid spreading to form a lamella. The authors do not mention the formation of the counterjet at all – probably because the height of impact is very small < 1 cm. In order to validate the model, the authors should explore the impact dynamics of hollow droplets when the height of impact is increased.
- The authors mention that with an increase in the volume fraction of air in the hollow droplet, the retraction velocity increases, and the coefficient of restitution decreases. Can the authors explain the physical significance of this behavior?
- The presence of a bubble increases the initial impact diameter of the bubble. How is the spreading radius defined?
- There is too much reference to the SI – the details of the relation between Weber number and encapsulated air are important and can be included in the main document.
- Line 130 – phrases such as kinetic energy of HD has a high positive value' are confusing and should be modified
- Line 150 – how is the inner surface area calculated? Do the authors take into account the distortion of the shape due to refraction at the liquid shell?

- Line 158 – what is effective surface tension? Effective surface tension going to zero can be misleading.
- What is the energy conversion argument?
- In the supplementary video, different surfactant concentrations are reported to be used for different experimental data sets. What is the rationale? Surfactants are supposed to play a major role in the impact dynamics – especially in the case of hollow droplets with multiple air-liquid interfaces.
- The hollow droplet does bounce from the substrate even when the height of impact is 10 mm. Therefore, the title of the paper – ‘non-rebound of hollow droplets’ can be misleading.

Response to the Comments by Reviewer #1

In this article, Zhou *et al.* reports an intriguing idea for inhibiting drops from bouncing off very liquid-repellent substrates. By including a large air bubble inside a water droplet, its jumping can be strongly suppressed on surfaces that are well-known for their repellent properties. I find this work nicely thought provoking as we often try to make superhydrophobic surfaces as repellent as possible. For example, the scientific discussion is often on how to further reduce the contact time between a bouncing drop and the substrate. Here, Zhou *et al.* argue for the need of a non-rebound (i.e., “infinite” contact time) yet highly liquid-repellent (i.e., low friction) substrate, since this will improve, e.g., the self-cleaning properties when keeping the impinging water drops on the sample while simultaneously allowing them to easily slide off. The inclusion of an air drop in the water drop is a beautifully simple solution to this problem.

In my opinion, the paper is very well written, interesting, and easy to read. The experiments have been carefully conducted with beautiful high-speed movies and the authors make a strong effort in discussing the possible reasons to why hollow drops do not bounce. Below, I list minor comments on how to further improve this work. After these changes, I warmly recommend this paper for publication in *Nat. Commun.* I believe the aesthetics of the experiments combined with the quite general system of a drop bouncing (or not) off a surface will attract a broad audience. Furthermore, the findings are important for the fluid dynamics and surface science community working on superhydrophobic surfaces, wetting, and drop dynamics.

Response: We express our gratitude to the referee for his/her recognition of our work and for recommending its publication. We also extend our sincere appreciation for providing us with valuable feedback and comments on our research paper. We have diligently reviewed the comments and have made the necessary revisions to improve the clarity of our work.

Minor comments

1. Regarding the surface tension.

- In the Methods section, you mention that the concentration of SDBS was “up to 0.2 wt%”. What does “up to” mean? Was the surface tension kept constant in all experiments or was there a big variation in the surface tension in different bouncing experiments? If the concentration is different in different experiments, the exact surface tension values should be mentioned in the captions of all figures and discussed in detail in the Methods section.

Response: We used SDBS solutions with varying concentrations to demonstrate the universal nature of hollow droplet (HD) rebound suppression across liquids with different properties, such as surface tensions. In addition to pure water, we employed three concentrations of SDBS solutions: 0.003 wt%, 0.1 wt%, and 0.2 wt% with surface tensions of 53 mN/m, 43 mN/m, and 35 mN/m, respectively. These experiments also allowed us to validate our theoretical models using data obtained from impact of droplets with different surface tensions.

In response to the referee’s comment, we have included the value of the liquid surface tension in the captions of Figures 1 and 2. Furthermore, within the ‘Methods’ section, we have provided a detailed description of the liquids used (Lines 277-285, Page 12).

- Why do you show profiles of drops with *different* SDBS concentrations in Supplementary Fig. 2b? Did a 0.2 wt% drop not bead up on a superhydrophobic surface?

Response: Supplementary Figure 2b was utilized to demonstrate that all liquid droplets exhibited a super-repellent state on their corresponding surfaces. Due to the varying surface tensions of the liquids, different solid surfaces were employed in our experiments to achieve the super-repellent state for each specific liquid. For instance, superhydrophobic surfaces were utilized for droplets of pure water and 0.003 wt% SDBS solution, while superamphiphobic surfaces were employed for droplets of 0.2 wt% SDBS solution and *n*-hexadecane. The low surface tension of the 0.2 wt% SDBS droplets caused them to wet the superhydrophobic surface, preventing them from beading up on the surface. Hence, superamphiphobic surfaces were utilized for 0.2 wt% SDBS droplets.

To address this comment appropriately, we have incorporated a detailed description of the

liquids and solid surfaces employed within the ‘Methods’ section (Lines 277-282, Page 12).

- Was SDBS added also to the SD water drops in the bouncing experiments of the paper? Please specify this in the main text.

Response: Yes, to ensure a fair comparison in our experiments, the liquid material of both the hollow droplet (HD) and single-phase droplet (SD) was kept consistent. In Figure 1, we presented a comparison of the rebound behavior between the HD and SD formed using a 0.003 wt% SDBS solution with a surface tension of 53 mN/m. To address the mentioned comment, we have explicitly stated this information in Line 65, Page 3 of our revised manuscript.

- Please mention the range of surface tensions used (35-?? mN/m) in the main text (Lines 55-56, p. 3) so that the reader understands that this is significantly below that of water.

Response: Following this this comment, we have clarified the range of surface tension (35-53 mN/m) in Lines 74-75, Page 3.

2. I cannot easily follow how you reached the expression for $\gamma_e \sim (R_0 - R_b)\gamma/R_0$ on Line 157 (p. 7). Since this is an important result in describing why the effective surface tension can be considered much smaller in your system, its derivation would need a more extensive explanation.

Response: In response to the referee’s comment, we have included a more comprehensive discussion on the derivation of the apparent surface tension (referred to as “effective surface tension” in the original manuscript) in Lines 219-229, Pages 9-10.

It is crucial to acknowledge that the estimation offers a qualitative analogy rather than an analytical solution for the apparent surface tension, $\gamma_a \sim (1 - \Phi^{1/3})\gamma$. The purpose of this estimation was to provide insights into the overall weakening of capillary effects resulting from the presence of an inner bubble within the droplets.

3. I assume that the pre-factors C in Eqs. 2, 3 and 4 are not to be considered the same? If this is correct, I would suggest giving these constants different variable names (e.g., A , B , and C or C_1 , C_2 and C_3) to avoid confusion.

Response: In line with this comment, we have revised the pre-factors C into C_1 , C_2 , C_3 and C_4 in Eqs. (1), (2), (3), and (4), respectively. This revision clarifies that these pre-factors have distinct values and should not be considered identical, thus avoiding any potential confusion.

4. Reg. Fig 4:

- This is a beautiful graph! Can you please mention in the caption that the dashed line is a **fit** of the theoretical prediction of Eq. 4.
- You write “ N indicates the time of droplet rebound.”. Do you mean “ N indicates the number of droplet rebounds.”?

Response: In accordance with the referee’s comments, we have revised the figure caption in Lines 511-513, Page 26, as follows: “The dashed line presents the theoretical prediction of the boundary between the two regimes, as calculated using Eq. (4) with a fitting prefactor of $C_4 = 1/7$. N indicates the number of droplet rebounds.”

5. In the discussion, you claim that you can “suppress the rebound, without altering the droplet composition or surface properties”. However, you do significantly change the surface tension of water by adding the stabilising surfactant (SDBS, see my comment in 1 above). I would recommend rephrasing this sentence.

Response: In agreement with the referee’s observation, we acknowledge that SDBS solution was predominantly used in our experiments as we utilized SDBS as a surfactant to stabilize the bubble. However, it is important to note the following points:

- (1) The suppression of rebound was demonstrated by comparing HDs to SDs with identical

mass, liquid material, and surface tension.

- (2) This phenomenon was observed for various liquids, including water with different SDBS concentrations and oil (*n*-hexadecane), indicating its applicability across different droplet compositions.

To address the vague point raised, we created the HD consisting of an air bubble encapsulated in pure water and repeated the impingement experiment. The suppression of rebound was also observed in the pure-water HD without altering the surface tension of water. We have included a video demonstration in Supplementary Movie 2.

6. I find the final comparison with compound balloons delightful!

Response: We thank the referee for acknowledging the comparison with compound balloons in our research. Our intention was to employ this analogy as a means to illustrate the concept and facilitate a better understanding of our findings.

7. A similar system of hollow droplets impacting a solid substrate has been studied before (see papers below). Although these papers report on the spreading dynamics and not rebound inhibition, I think a reference to their work would be appropriate.

- “Hollow droplets impacting onto a solid surface”, I. P. Gulyaev & O. P. Solonenko, *Experiments in Fluids* 54, 1432 (2013). <https://doi.org/10.1007/s00348-012-1432-z>
- “Impact dynamics of air-in-liquid compound droplets”, D. P. Naidu and S. Dash, *Physics of Fluids* 34, 073604 (2022), <https://doi.org/10.1063/5.0096599>

Response: In accordance with the referee’s comments, we have incorporated the suggested papers as References 21 and 32 in our revised manuscript. Furthermore, we have included an additional paragraph in the ‘Introduction’ section (Lines 44-54, Pages 2-3) to provide a more comprehensive research background on HDs.

Response to the Comments by Reviewer #2

The manuscript concerns a study about droplets impacting on superhydrophobic surfaces. In particular, the inclusion of a bubble inside the droplet, as the authors claim, suppresses the rebound phenomenon observed when a droplet impacts on a superhydrophobic surface. The suppression of the rebound is interesting however the study seems premature, in the sense that the analysis is not clear and does not give a picture of what really happens.

Response: We express our gratitude to the referee for providing valuable comments that have significantly contributed to clarifying the vague points in the original manuscript. The feedback has led to substantial improvements in the revised manuscript, allowing for a more detailed and comprehensive understanding of the phenomenon under investigation. We have diligently revised and enhanced the manuscript, incorporating in-depth explanations of theoretical models, conducting new simulations, and conducting additional experiments to address the raised points. These revisions have resulted in a more robust and informative presentation of our research findings.

--For example, the inclusion of the bubbles notably changes the apparent density of the composite droplet. And when the Weber number is used to compare the single and the composite droplet, which density is used to calculate We ?

Response: We acknowledge the referee's comment regarding the influence of the inclusion of a bubble on the apparent density of the hollow droplet. Upon careful consideration, we have concluded that utilizing the liquid density, rather than the apparent density, is the most appropriate approach to define the Weber number in our study. To clarify this point, we have included a detailed discussion in the 'Methods' section (Lines 325-335, Page 14).

--Also, in figure 1f the non-rebound is observed when the ratio, liquid/bubble volume, Φ , > 0.4 . At this magnitude Φ , the composite system is completely different than the one of the single

liquids. I mean that the systems are not comparable. We could mostly term it bubble and not droplet with a small bubble inside it.

Response: We extend our sincere appreciation to the referee for raising this question and providing your perspective. We understand the concern regarding the presence of a bubble and its impact on the fluid system structure. However, we would like to highlight that the comparison between hollow droplets (HDs) and single-phase droplets (SDs) remains relevant and informative based on the following considerations:

Firstly, the objective of our study is to investigate the impact behavior of HDs on super-repellent surfaces and the suppression of rebound. It is worth noting that numerous studies in the literature have conducted comparative investigations between HDs and SDs, highlighting the distinct characteristics and behavior of HDs (References 1-4).

Secondly, we have taken meticulous care in controlling the mass, liquid material, and release height of both HDs and SDs to ensure a direct comparison between the two systems. This careful control ensures that any observed differences can be attributed to the presence of the bubble in HDs.

Lastly, and most importantly, our research primarily focuses on the bouncing behavior of droplets on super-repellent surfaces, a behavior observed for both SDs and HDs upon impact. The characteristic parameters such as contact time, maximum spreading radius, contact radius, retraction velocity, and restitution coefficient are essential for describing this behavior and can be measured and compared between the two systems. Therefore, in terms of droplet impact and bouncing behavior, the two systems are directly comparable. It is important to note that for the specific case of droplet rebound studied in our research, SDs can be considered as a special instance of HDs with a fully filled core ($\Phi = 0$).

To address the referee's concern and provide a more comprehensive literature review, we have added a paragraph to the revised manuscript in the 'Introduction' section (Lines 44-54, Pages 2-3). This addition will further emphasize the uniqueness of the rebound suppression behavior and enrich the discussion of HDs in our paper.

References

1. Gulyaev, I. P. & Solonenko, O. P. Hollow droplets impacting onto a solid surface. *Exp.*

Fluids 54, 1432 (2012).

2. Li, D., Duan, X., Zheng, Z. & Liu, Y. Dynamics and heat transfer of a hollow droplet impact on a wetted solid surface. *Int. J. Heat Mass Transf.* 122, 1014-1023 (2018).
3. Naidu, D. P. & Dash, S. Impact dynamics of air-in-liquid compound droplets. *Phys. Fluids* 34, 073604 (2022).
4. Nasiri, M., Amini, G., Moreau, C. & Dolatabadi, A. Flattening of a hollow droplet impacting a solid surface. *J. Fluid Mech.* 962, A1 (2023).

--The analysis in page 6 is not convincing. The argument that although the energy dissipation is very small but the velocity distribution, in the composite system, is such that rebound is suppressed is not solid.

Response: To address this concern, we have conducted additional simulations at a higher impact velocity ($V_0 = 0.270$ m/s) in addition to the one included in the original manuscript ($V_0 = 0.135$ m/s). The results of these simulations, along with the corresponding analysis, have been added to the revised manuscript (Figures 3, 4, and 5, Supplementary Figure 7, Supplementary Movie 4, and Lines 175-191, Page 8).

In the newly performed simulations, the higher impact velocity ($V_0 = 0.270$ m/s) resulted in the rebound of the HD but with a suppressed rebound height compared to the SD. This finding highlights the rich dynamics of HDs and complements the simulation of non-rebounding HDs at a lower impact velocity ($V_0 = 0.135$ m/s). Both simulations ($V_0 = 0.135$ m/s and $V_0 = 0.270$ m/s) indicate that HDs and SDs follow different energy conversion pathways. Specifically, HDs exhibit a low conversion efficiency from kinetic energy to surface energy (Figure 3) and experience velocity oscillations during the retraction stage (Figure 4). The simulations suggest that these velocity oscillations significantly attenuate the net momentum of HDs, resulting in either a negative net momentum (Supplementary Figure 7c) for non-rebounding or a low positive value (Supplementary Figure 7d) that causes the HD to rebound with a suppressed height. This behavior is distinct from that of SDs, where higher rebound is associated with a larger net momentum (Figure 4).

In our study, we conducted a comprehensive investigation using a combination of experimental observations and numerical simulations to ensure the robustness and reliability of our conclusions. The agreement between the experimental results and the numerical simulations reinforces the consistency and accuracy of our findings. The addition of these new simulations and their analysis strengthens the comprehensiveness of our study and provides further insights into the dynamics of HDs at different impact velocities.

The importance of using a relatively big bubble inside a droplet suffers from the fact that what suppression is only observed when gravity is present, i.e. in free fall. What happens in the case of a impact in side wall. Then the position of this big bubble could unpredictably change.

Response: We would like to clarify that in our experiments, we ensured that the effect of gravity was negligible by keeping the $R_h < l_c$ (capillary length). Thus, the rebound suppression observed in our study is not caused by gravity but is instead a result of the counteractive capillary effect, as extensively discussed in the subsection titled “Mechanism for rebound suppression” (Lines 172-235, Pages 8-10).

The presence of the internal bubble in HDs disturbs the consistency of the velocity flow field, and the counteractive change of the inner and outer surface areas hinders the conversion of surface energy back to kinetic energy, preventing effective take-off of HDs. This mechanism is the primary reason for the suppressed rebound observed in our experiments.

To address the concern raised by the referee regarding the impact on a side wall, we have conducted additional experiments to demonstrate that rebound suppression still occurs when HDs impact an inclined surface in the Leidenfrost state. Furthermore, we have performed new experiments to compare the impact of SDs and HDs on vertical superamphiphobic surfaces. In these experiments, the SD bounces off the surface while the HD slides down along the surface, clearly indicating the suppressed rebound of the HD. These additional experiments, which are presented in Supplementary Movie 2, further support our understanding that gravity plays no significant role in suppressing the rebound of HDs.

Response to the Comments by Reviewer #3

The article by Zhou et al. approaches the classical problem of bouncing of water drops on a water-repellent surface, but with a twist: the drop in this case has a bubble encapsulated within. The authors compare the bouncing dynamics in the two cases to conclude that bubble inclusions may be utilized to trigger suppression of bouncing, without enhancing dissipation. The authors map this effect over Weber numbers varying over three orders of magnitude and supplement their arguments further with numerical simulations.

The work is (mostly) adequate, so is the writing (I also appreciate the video of the balloon). However, the observations in this case are perhaps better suited in a droplet-specific journal. I would just add a couple of remarks, which the authors could consider for their future endeavors with this work.

Response: We express our sincere gratitude to the referee for their positive comments on our study regarding the bouncing dynamics of hollow droplets (HDs) on super-repellent surfaces. We are grateful for the referee's valuable remarks, as they have further enhanced our analysis and interpretation of the findings presented in the revised manuscript. We are confident that the revised version of the manuscript provides a more comprehensive and insightful understanding of the phenomenon.

First, I think it would be wise to plot the actual contact time values, which the authors do not plot anywhere.

Response: In response to this feedback, we have revised Figure 1f by plotting the actual contact time values against the bubble volume fraction (see Figure R1 below).

Figure R1 (Figure 1f in revised manuscript). **Dependence of the contact time t_c on bubble volume fraction Φ .** $\Phi = 0$ stands for SD while $\Phi > 0$ for HD. In the case of the non-rebound HD, the contact time is set to be equal to the droplet's complete evaporation time t^* , which was approximately 7,200 s in experiments. Insets, snapshots of a bouncing HD with $\Phi = 0.2$ (left) and a resting HD with $\Phi = 0.8$. Scale bar, 1 mm.

To compare it with the original plot of Richard et al. which reported the Hertzian nature of the bouncing dynamics for the first time, they could plot the contact time against V for different ϕ (the ratio of the bubble to total volume). If dissipation is indeed negligible, as the authors claim, then they should see flat horizontal lines for different ϕ and even recover the line of Richard et al. for $\phi = 0$. Similarly, the authors should plot the contact time against the radius on a log-log plot to check if the scaling is indeed $3/2$.

Response: In accordance with this comment, we have conducted additional experiments to compare the contact times of HDs and SDs. These experiments align with the study conducted by Richard *et al.* The outcomes of these experiments have been visually presented in Supplementary Figure 5 and extensively discussed in Lines 100-107, Pages 4-5.

Figure R2 (Supplementary Figure 5 in revised manuscript) **Contact time of impacting droplets.** **a** The contact time as a function of impact velocity V_0 for different Φ . $R_h = 0.89$ mm. **b** The contact time against the apparent radius R_h . $V_0 = 0.65$ m s^{-1} . **c** Comparison of the experimental contact time t_c against theoretical inertial-capillary timescale τ . Data was obtained by using droplets (35 mN m^{-1}) to impact the superamphiphobic surface.

Remarkably, we observed a reduction in the contact time of bouncing HDs as compared to that of bouncing SDs, under equivalent conditions. This reduction can be attributed to the smaller spreading factor and faster retraction velocity exhibited by HDs, as demonstrated in Supplementary Figure 6. Despite the shorter contact time, we have demonstrated that the rebound of HDs remains suppressed in comparison to SDs. This is substantiated by the diminished rebound height and, consequently, the smaller restitution coefficient of HDs, as depicted in Supplementary Figure 6c.

Finally, for the case presented by the authors, as we increase f for a given R_0 , we decrease the volume of water, yet increase the contact time. This is in sharp contrast to droplets impinging

on superhydrophobic macrotextures (like lines or points), where the volume is diminished too (via reorganization) leading to a decrease in contact time (Bird et al. *Nature* 2013 and Gauthier et al. *Nat. Commun.* 2015). This needs to be discussed – why decreasing volume does not decrease the contact time for the author’s case, as one would expect from Quéré’s inertia-capillary scaling (Richard et al. *Nature* 2002). The increase in the contact time could be quantified as a function of ϕ to elaborate on this.

Response: Through our additional experiments, we have made an intriguing observation regarding the contact times (t_c) of non-rebound HDs and bouncing HDs. Notably, we found that these two cases exhibit distinct contact times. In the case of non-rebound HDs, where the droplets remain in contact with the solid surface after impingement, the contact time can be assumed to tend towards infinity when compared to the bouncing scenario, under the assumption of negligible droplet evaporation. However, contrary to this expectation, we discovered that the contact time for bouncing HDs is even smaller than that of SDs, as illustrated in Figure R2. For instance, we observed $t_c = 2.23\tau$ for bouncing SDs, while $t_c = 1.74\tau$ for bouncing HDs (Figure R2c). It is worth noting that the pre-factor of 2.23 for SDs aligns with the findings of Bird *et al.* published in *Nature* 2013.

It is important to emphasize that this reduction in contact time is not due to a difference in droplet mass or volume since, in our comparison, the mass of HDs and SDs remained consistent. Rather, the reduced contact time can be attributed to the diminished spreading and accelerated retraction of HDs, as evidenced by their smaller spreading factor and faster retraction velocity (Supplementary Figure 6), respectively.

Furthermore, it is essential to note that our current study primarily focuses on investigating the dynamics of suppressed rebound in HDs. The contact time of bouncing HDs, which exhibits a distinct behavior, falls beyond the scope of our present work. Therefore, a comprehensive and independent investigation of the contact time in bouncing HDs is warranted and merits further exploration.

Response to the Comments by Reviewer #4

The paper presents an experimental study on drop impact on phobic surfaces, showing how the presence of bubbles can promote rebound suppression.

Although the effect is limited to a relatively narrow region of We numbers (see Fig. 4), with the non-rebound to rebound threshold changing from ~ 0.1 to ~ 1 , the effect appear to be consistent and observed also in other conditions, such as Leidenfrost boiling, as well as other macroscopic systems, such as water balloons.

I also appreciate that the authors have included a simple spring model, based on a two-spring system, that nonetheless seems to reproduce well the experimental data.

I have no comments on the paper, and I recommend its publication.

Response: We express our gratitude to the referee for providing positive feedback and recommending our work for publication.

Response to the Comments by Reviewer #5

The manuscript reports differences in the impact behavior of a simple and an air-encapsulated hollow droplet on superhydrophobic substrates.

Response: We extend our gratitude to the referee for his/her constructive comments and insightful suggestions. We acknowledge that the feedback has been instrumental in clarifying potential ambiguities and refining the content of our revised manuscript. We greatly appreciate his/her expertise and careful evaluation, as it has undoubtedly contributed to the overall improvement of our work.

There have been several papers over the last 2-3 years that have reported experimental observations and numerical analysis of impact of hollow droplets on different substrates

1. Nasiri, M., Amini, G., Moreau, C. and Dolatabadi, A., 2023. Flattening of a hollow droplet impacting a solid surface. *Journal of Fluid Mechanics*, 962, p.A1.
2. Nasiri, M., Amini, G., Moreau, C. and Dolatabadi, A., 2021. Hollow droplet impact on a solid surface. *International Journal of Multiphase Flow*, 143, p.103740
3. Naidu, D.P. and Dash, S., 2022. Impact dynamics of air-in-liquid compound droplets. *Physics of Fluids*, 34(7), p.073604.
4. Wei, Y. and Thoraval, M.J., 2021. Maximum spreading of an impacting air-in-liquid compound drop. *Physics of Fluids*, 33(6), p.061703.

Please note that this is not the comprehensive list of papers and there are several more. There are no citations to these papers in the manuscript. I am not sure if the authors are aware of these works.

Response: In response to this comment, we have incorporated a new paragraph into the 'Introduction' section of our manuscript, focusing on providing a comprehensive research background of hollow droplets (HDs), in Lines 44-54, Pages 2-3. We have appropriately cited the four mentioned papers, namely References 29, 31, 32, and 28, among others, to ensure

proper acknowledgment of their contributions.

•During impact of a hollow droplet on a substrate, a counterjet forms in addition to the liquid spreading to form a lamella. The authors do not mention the formation of the counterjet at all – probably because the height of impact is very small < 1 cm. In order to validate the model, the authors should explore the impact dynamics of hollow droplets when the height of impact is increased.

Response: Indeed, it is correct that a counter-jet is observed at a high impact height, as depicted in Figure R3, where the impact height is set at 100 mm. However, unlike HDs impacting hydrophilic surfaces, where the counter-jet typically exhibits a continuous upward velocity and eventually leads to bubble breakup, our observations reveal only the initial development of a dimple-shaped counter-jet (Figures R3 and R4) when HDs impinge on super-repellent surfaces. Therefore, we have introduced additional discussions in Lines 206-209 on Page 9 to highlight this difference.

We deeply appreciate the referee’s suggestion for optimizing our model. In our experimental setup, the maximum impact height we utilized was 150 mm, corresponding to a moderate to large Weber number ($We \approx 114$). The predicted maximum spreading radius of HDs demonstrated good consistency at this height (Figure 2c in the revised manuscript). However, as the impact height exceeds 10 mm, both HDs and SDs exhibit unavoidable splashing during impact, resulting in a decrease in the droplet’s mass after rebound (Figure R3). Consequently, we purposely controlled the impact height to remain below 10 mm to ensure the integrity of both HDs and SDs in validating our prediction of the restitution coefficient.

Figure R3 HD impact surface at a high impact height. Release height:100 mm. Scale bar, 1mm.

Figure R4 Dimple-shaped surface (Figure 5a in revised manuscript). Schematic (left) and pictures displaying the asynchronous deformation of the inner and outer surfaces, which induces capillary flows opposing each other. The dashed yellow boxes highlight the dynamic changes of the dimple shape on the inner surface. Scale bar, 1mm.

•The authors mention that with an increase in the volume fraction of air in the hollow droplet, the retraction velocity increases, and the coefficient of restitution decreases. Can the authors explain the physical significance of this behavior?

Response: The behavior of HDs modified by the presence of bubbles holds significant physical significance in two main aspects.

Firstly, the inclusion of a bubble within the HD leads to a reduction in the lamella thickness (h_0) of the droplet during retraction. This reduction can be described by the relationship $h_0 \sim R_0 We^{-1/2} (1 - \Phi^{1/3})^{1/2}$ based on our model. Consequently, this reduction amplifies the capillary pressure ($\sim \gamma/h_0$) that drives the droplet's retraction. As a result, the retraction velocity of HDs, denoted as V_{ret} , increases compared to SDs with no bubble present. Specifically, the retraction velocity is given by $V_{\text{ret}} = C_2 \left(\frac{\gamma}{\rho R_0} \right)^{1/2} We^{1/4} \left[1 - \Phi^{1/3} \right]^{-1/4}$, where C_2 is a constant and Φ represents the bubble volume fraction.

Secondly, the presence of the bubble within the HD disrupts the internal flow field of the droplet, leading to a decrease in the net upward momentum during rebound. This reduction results in a suppressed rebound height and, consequently, a lower restitution coefficient. In our study, we provide a quantitative estimation of this reduced restitution coefficient, denoted as ε , given by $\varepsilon = C_3 We^{-1/2} \left[1 - We^{-1/2} \Phi^{2/3} \left(A - \Phi A^{-1/2} \right)^{-1} \right]$, where C_3 is a constant and A represents

a dimensionless coefficient.

It is worth noting that larger bubbles, characterized by a higher value of Φ , generate higher capillary pressures during the retraction stage and exert a more pronounced weakening effect on the net upward momentum during rebound. In practical terms, the inclusion of a gas bubble within the droplet enables HDs to retract at a faster rate, resulting in a reduced contact time compared to SDs. Additionally, HDs with bubbles also exhibit relatively lower bouncing heights during rebound.

Overall, these findings highlight the profound impact of bubbles on the dynamics of HDs, influencing their retraction behavior, contact time, and rebound characteristics.

•The presence of a bubble increases the initial impact diameter of the bubble. How is the spreading radius defined?

Response: In our study, the maximum spreading radius R_{\max} of the HD was defined as the apparent radius of the liquid rim at the state of maximum spreading, as illustrated in Figure R5.

Figure R5 Illustration of the maximum spreading radius (Figure 2a in revised manuscript).

There is too much reference to the SI – the details of the relation between Weber number and encapsulated air are important and can be included in the main document.

Response: In response to this suggestion, we have revised the manuscript by including the details of the relation between Weber number and encapsulated air bubble in the subsection entitled ‘Suppressed rebound of hollow droplets’ (Lines 122-130, 132-135, 144-156, and 163-

166, Pages 5-7).

•Line 130 – phrases such as kinetic energy of HD has a high positive value’ are confusing and should be modified

Response: In line with this comment, we have revised the statement into “the non-rebound HDs maintain a high and nearly constant value of kinetic energy after impact” (Lines 186-187. Page 8).

•Line 150 – how is the inner surface area calculated? Do the authors take into account the distortion of the shape due to refraction at the liquid shell?

Response: The inner, outer, and whole surface areas were derived from our numerical simulations rather than from experimental results. Therefore, they are not subject to the limitations inherent in physical measurement procedures. In our simulations, we employed a two-dimensional, axis-symmetric setup. Consequently, the surface areas were calculated by numerically integrating over the curve that represents the surface of the droplet.

•Line 158 – what is effective surface tension? Effective surface tension going to zero can be misleading.

Response: To clarify the vague point, we have changed the term “effective surface tension” to “apparent surface tension” in the revised manuscript. The apparent surface tension (γ_a) is used to represent the net capillary effect resulting from the counteractive capillary flows between the inner and outer gas-liquid surfaces of HDs. Due to the opposing nature of these surfaces, the apparent surface tension can be approximated as the difference in capillarity between the two surfaces, $R_0\gamma_a \sim (R_0\gamma - R_b\gamma)$, which results in $\gamma_a \sim (R_0 - R_b)\gamma/R_0 = (1 - \Phi^{1/3})\gamma$. This expression proves to be valuable in comprehending HDs modeled as a two-spring system with reduced elasticity.

The physical significance of γ_a approaching zero lies in the cancellation of capillary effects between the inner and outer surfaces. This cancellation implies that any decrease (or increase) in the energy of the outer surface is entirely compensated by an increase (or decrease) in the energy of the inner surface. Consequently, the conversion efficiency from surface energy to kinetic energy approaches zero. In this scenario, the changes in the inner and outer surface areas become out of phase during droplet retraction (Figures 5c and 5d in the revised manuscript).

We have revised the discussion on the apparent surface tension in Lines 219-229, Pages 9-10.

- What is the energy conversion argument?

Response: HDs and SDs exhibit distinctive energy conversion pathways. During droplet spreading, almost all the kinetic energy in SDs is converted into surface energy. Consequently, during droplet retraction, the surface energy is transformed back into kinetic energy, resulting in the rebound of SDs. This rebound behavior is a direct consequence of the efficient energy conversion process in SDs. On the other hand, in the case of HDs, a smaller proportion of the kinetic energy is transferred into surface energy during spreading. Moreover, an additional energy conversion mechanism occurs between the inner and outer surfaces of the HDs. This additional conversion process reduces the overall efficiency of converting surface energy back into kinetic energy during retraction in HDs.

To reflect these differences, we have revised the energy conversion argument in Lines 182-191 on Page 8, specifically addressing droplet spreading, as well as in Lines 219-223 on Pages 9-10, focusing on droplet retraction.

- In the supplementary video, different surfactant concentrations are reported to be used for different experimental data sets. What is the rationale? Surfactants are supposed to play a major role in the impact dynamics – especially in the case of hollow droplets with multiple air-liquid interfaces.

Response: Different surfactant concentrations were employed in this study for two specific reasons. Firstly, the surfactant was utilized to stabilize the encapsulated bubble within the droplet. Secondly, by altering the surfactant concentrations, the surface tension of the liquid could be modified. This variation in surface tension was instrumental in demonstrating that the suppression of HD rebound is a universal phenomenon, independent of the specific materials used.

Notably, our research findings indicated that the impact of surfactants on rebound suppression can be considered negligible. This conclusion was drawn based on our observations of the same rebound suppression phenomenon using HDs made from pure water, SDBS HDs in the Leidenfrost regime, and numerical simulations of droplet impact that solely accounted for the liquid's surface tension without incorporating the presence of a surfactant. This empirical evidence led us to assert that understanding the rebound suppression phenomenon can be achieved from a hydrodynamic perspective, rather than relying on molecular dynamics.

To address any potential ambiguity, we have revised Supplementary Movie 2 to showcase the impact of the pure-water HD. Additionally, we have provided a detailed description of the liquids used in the 'Methods' section in Lines 277-285 on Page 12.

- The hollow droplet does bounce from the substrate even when the height of impact is 10 mm. Therefore, the title of the paper – 'non-rebound of hollow droplets' can be misleading.

Response: In accordance with the referee's comment, we have revised title into "**Suppression of hollow droplet rebound on super-repellent surfaces**".

REVIEWERS' COMMENTS

Reviewer #1 (Remarks to the Author):

The authors have made all of my suggested changes. In my opinion, they have also responded well to the questions and comments by the other reviewers. I recommend the paper for publication in Nature Communications.

Reviewer #2 (Remarks to the Author):

The authors made a considerable effort to address the comments of the reviewers.

In what concerns however my comments I would mention the following:

a) The selection of the liquid density for calculating the Weber number is misleading although the authors claim that this is the best selection. And it is misleading because the system is not a liquid mass. It is composite with a big part of air i.e. a bubble with some liquid on the lower part. And the selection of the liquid density for the Weber number shifts the velocity regions to lower impact velocities, i.e favoring non rebound.

b) concerning the impact of gravity and the impact on side walls I meant:

that gravity forces (makes) the liquid part of the HD to be at the bottom. I didn't mean anything about the capillary length and its comparison with the H droplet radius. If gravity is not present, then the droplet may impact the solid surface by having the liquid part on the sides. And that is why I made the comment on impacting side walls and NOT inclined walls.

Based on the above but I can't suggest publication of this work in Nat Comms. since it doesn't meet the quality criteria of the journal.

I would suggest publication in a more droplets-oriented journal as another reviewer suggested.

Reviewer #3 (Remarks to the Author):

The authors have responded reasonably to my concerns. However, I must add that to first make a surface super-repellent (such that it minimizes the contact time of an impacting drop) and then to investigate means to maximize the contact time (inclusion of a bubble) is a rather esoteric problem, and perhaps not as 'grand' as the authors claim in the very introduction itself: 'the non-rebound behavior of capillary-dominant droplets on super-repellent surfaces poses a grand challenge...'.

The work, however, is adequate in itself. So, I recommend the article for publication with the strong recommendation against using the phrase 'grand challenge'.

Reviewer #5 (Remarks to the Author):

The authors have responded to my previous comments.

Response to the Comments by Reviewer #1

The authors have made all of my suggested changes. In my opinion, they have also responded well to the questions and comments by the other reviewers. I recommend the paper for publication in *Nature Communications*.

Response: We thank the referee very much for his/her positive feedback and for recommending our manuscript for publication in *Nature Communications*.

Response to the Comments by Reviewer #2

The authors made a considerable effort to address the comments of the reviewers. In what concerns however my comments I would mention the following:

a) The selection of the liquid density for calculating the Weber number is misleading although the authors claim that this is the best selection. And it is misleading because the system is not a liquid mass. It is composite with a big part of air i.e. a bubble with some liquid on the lower part. And the selection of the liquid density for the Weber number shifts the velocity regions to lower impact velocities, i.e favoring non rebound.

Response: We thank the referee for the invaluable input and evaluation of our study. We acknowledge the reviewer's apprehension regarding the utilization of liquid density in defining the Weber number (We). The hollow droplet (HD) encompasses one or more bubbles within a liquid shell, forming a composite system distinct from a single-phase droplet (SD). Consequently, the parameters utilized to describe such a composite system must necessarily differ. We emphasize that a comprehensive understanding of the composite system's characteristics can only be achieved by concurrently employing the bubble volume fraction (Φ) and the Weber number (We), as elucidated in our model and Equations (1)-(4) in the manuscript. Sole utilization of We would be inadequate in modeling the rebound behavior of the composite system, potentially leading to erroneous conclusions.

Remarkably, the effective density-based Weber number (We_a) can be expressed as a function of We and Φ , $We_a = We(1 - \Phi)$. Should we substitute We with We_a in our study, it would influence solely the structure of the developed equations, leaving the accuracy and predictive capability of our model unaltered. As an example, for the prediction of the spreading factor, the utilization of We_a would transform Equation (1) into Equation (1'):

$$\beta = C_1 We_a^{\frac{1}{4}} \left[1 - \Phi \left(1 + \Phi^{-\frac{1}{3}} We_a^{-\frac{1}{2}} \right)^{-\frac{3}{2}} \right]^{\frac{1}{2}}, \quad (1)$$

$$\beta' = C_1 We_a^{\frac{1}{4}} (1 - \Phi)^{-\frac{1}{4}} \left[1 - \Phi \left(1 + \Phi^{-\frac{1}{3}} (1 - \Phi)^{\frac{1}{2}} We_a^{-\frac{1}{2}} \right)^{-\frac{3}{2}} \right]^{\frac{1}{2}}. \quad (1')$$

The predictions derived from Equation (1) and (1') are presented in Figure R1, confirming the identity of β and β' .

Figure R1. Consistency in theoretical predictions using We or We_a . A comparative analysis between the experimental value and the theoretical predictions of the spreading factor calculated using (left) Equation (1) and (right) Equation (1').

Lastly, it is pertinent to note that all comparisons between SD and HD were executed utilizing droplets of identical mass and released from identical heights, ensuring equal impact velocity and kinetic energy. Notably, the suppression of rebound or non-rebound phenomena was observed exclusively in HD when compared to SD. This suppression effect is robust and independent of the Weber number definition.

b) concerning the impact of gravity and the impact on side walls I meant: that gravity forces (makes) the liquid part of the HD to be at the bottom. I didn't mean anything about the capillary length and its comparison with the H droplet radius. If gravity is not present, then the droplet

may impact the solid surface by having the liquid part on the sides. And that is why I made the comment on impacting side walls and NOT inclined walls.

Based on the above but I can't suggest publication of this work in *Nat Comms*. since it doesn't meet the quality criteria of the journal. I would suggest publication in a more droplets-oriented journal as another reviewer suggested.

Response: We genuinely appreciate the reviewer's remarks concerning the influence of liquid location and fully comprehend the concern that liquid location may affect the bouncing behavior of HDs. To address this concern, we conducted additional experiments, contrasting the behavior of HD and SD during under-liquid bouncing (Supplementary Movie 2). In this scenario, both HD and SD ascended within a continuous liquid and subsequently impacted a solid surface due to buoyancy forces. While SD rebounded upon impact, HD remained wobbling on the surface. Unlike airborne HD impacts, where the bottom liquid-rich portion first contacts the solid surface, under-liquid HD impact manifests with the top bubble-rich portion making initial contact. Furthermore, when examining the impact of HD on a side wall, both the liquid-rich and bubble-rich portions simultaneously come into contact with the solid wall (Figure R2). Remarkably, the suppression of rebound persists with HD usage in this particular context as well (Supplementary Movie 2). These experimental findings conclusively demonstrate that the liquid location at impact does not exert an influence on the suppression of rebound, thereby affirming the robustness and universality of this phenomenon.

Figure R2. Simultaneous contact of liquid-rich and bubble-rich portions at impact.

Response to the Comments by Reviewer #3

The authors have responded reasonably to my concerns. However, I must add that to first make a surface super-repellent (such that it minimizes the contact time of an impacting drop) and then to investigate means to maximize the contact time (inclusion of a bubble) is a rather esoteric problem, and perhaps not as ‘grand’ as the authors claim in the very introduction itself: ‘the non-rebound behavior of capillary-dominant droplets on super-repellent surfaces poses a grand challenge...’.

The work, however, is adequate in itself. So, I recommend the article for publication with the strong recommendation against using the phrase ‘grand challenge’.

Response: We express our deep gratitude to the referee for the valuable positive feedback and for endorsing our manuscript for publication in *Nature Communications*. In response, we have removed the term "grand" from the revised version of the manuscript.

Response to the Comments by Reviewer #5

The authors have responded to my previous comments.

Response: We express our deep gratitude to the referee for providing this feedback and for his/her input in reviewing this manuscript.